# Low-frequency conductivity of low wear high-entropy alloys

Cheng-Hsien Yeh [1], Wen-Dung Hsu [2,3,4] ✉, Bernard Haochih Liu [2,3] ✉, Chan-Shan Yang [3,5] ✉, Chen-Yun Kuan [2], Yuan-Chun Chang [2], Kai-Sheng Huang [2], Song-Syun Jhang [2], Chia-Yen Lu [5], Peter K. Liaw [6] & Chuan-Feng Shih [1,3,4] ✉

High-entropy alloys (HEAs) provide new research avenues for alloy combinations in the periodic table, opening numerous possibilities in novel-alloy applications. However, their electrical characteristics have been relatively underexplored. The challenge in establishing an HEA electrical conductivity model lies in the changes in electronic characteristics caused by lattice distortion and complexity of nanostructures. Here we show a low-frequency electrical conductivity model for the Nb-Mo-Ta-W HEA system. The cocktail effect is found to explain trends in electrical-conductivity changes in HEAs, while the magnitude of the reduction is understood by the calculated plasma frequency, free electron density, and measured relaxation time by terahertz spectroscopy. As a result, the refractory HEA $Nb_{15}Mo_{35}Ta_{15}W_{35}$ thin film exhibits both high hardness and excellent conductivity. This combination of $Nb_{15}Mo_{35}Ta_{15}W_{35}$ makes it suitable for applications in atomic force microscopy probe coating, significantly improving their wear resistance and atomic-scale image resolution.

The field of high-entropy alloys (HEAs) is advancing rapidly. The phase-stability effects of configurational entropy and the materials harmonization functionality brought about by the charming cocktail effect give rise to a greater variety of material combinations[1–4]. Previous research on HEAs has predominantly focused on their mechanical properties, while investigations into their electrical characteristics remain limited. Apart from their unique low-temperature superconducting properties[5], one of the most significant electrical features observed in the literature for HEAs is their temperature-independent conductivity[6]. Additionally, the direct-current (DC) conductivity of HEAs is generally orders of magnitude higher than that of pure metals or low-entropy alloys. This feature possibly associates with electron scattering caused by severe lattice distortion, resulting in lower electron mobility and conductivity in HEAs at room temperature, thus

reducing the temperature dependence of their conductivity properties. However, the conduction mechanisms in HEAs have yet to be explored, and alloys that balance both high conductivity and great mechanical properties have not been developed. Establishing the metallic conductivity necessitates considering the intrinsic electronic concentration, mobility, and relaxation time of the crystal-structure nature, while also considering extrinsic factors, such as phase compositions, nanostructures, and surface properties, all of which remain unexplored in the realm of HEAs. Moreover, the severe distortion phenomena in HEAs further complicate the establishment of a conductivity model[7].

Huang et al. reported that the addition of copper in the $AlCu_xNiTiZr_{0.75}$ alloy resulted in a significant reduction in electrical resistivity, with the lowest recorded at approximately 66 μΩ·cm when

[1]Department of Electrical Engineering, National Cheng Kung University, Tainan 70101, Taiwan. [2]Department of Materials Science and Engineering, National Cheng Kung University, Tainan 70101, Taiwan. [3]Applied High Entropy Technology (AHET) Center, National Cheng Kung University, Tainan 70101, Taiwan. [4]Program on Semiconductor Packaging and Testing, Academy of Innovative Semiconductor and Sustainable Manufacture, National Cheng Kung University, Tainan 70101, Taiwan. [5]Institute and Undergraduate Program of Electro-Optical Engineering, National Taiwan Normal University, Taipei 11677, Taiwan. [6]Department of Materials Science and Engineering, The University of Tennessee, Knoxville, TN 37996, USA. ✉e-mail: wendung@mail.ncku.edu.tw; hcliu@mail.ncku.edu.tw; csyang@ntnu.edu.tw; cfshih@mail.ncku.edu.tw

$x$ is 1.4, compared to 363 µΩ·cm at $x = 1$[8]. The authors qualitatively attributed this phenomenon to the presence of copper, which altered the binding energy and increased the electron-relaxation time, subsequently augmenting the density of states near the Fermi level. In 2010, Ye et al.[9] reported that the TiFeCoNi HEA thin films exhibited an low electrical resistivity of approximately 35 µΩ·cm after 1000 °C high-temperature annealing following oxide formation. In 2018, the same team conducted further nanostructure analysis and proposed that the material is a mixture of oxygen-deficient metal oxides and metals, with an uneven phase composition[10]. However, the research group did not delve deeper into the mechanisms underlying the observed low resistivity of high-entropy oxides, and further investigation is needed to clarify these mechanisms.

Besides the high-entropy oxide, the refractory HEA, Nb-Mo-Ta-W-Ti-Zr-Hf-V, with a body-centered-cubic (BCC) structure has recently garnered significant attention in the field of HEAs due to its high-temperature mechanical properties. For its electrical properties, Kim et al.[11] reported a resistivity of approximately 168 µΩ·cm for NbMoTaW thin films, demonstrating good hardness at around 12 GPa. Feng et al.[12] conducted a study on nanocrystalline NbMoTaW films, finding a resistivity of 170 µΩ·cm and a hardness of 16.0 GPa. They proposed a strong positive correlation between the conductivity and grain size, as well as thickness. These studies also indicate the stability of the resistivity of NbMoTaW at elevated temperatures. However, there is currently still a lack of in-depth research on the electrical conductivity mechanisms of BCC refractory HEAs. A simple model with roughly estimated stress coefficients and bulk resistivity has been used[12]. However, it can't explain the size-independent resistivity in BCC refractory HEA films. Furthermore, these resistivities still remain two orders of magnitude higher than that of pure metals and over an order of magnitude greater than that of binary alloys.

The term, "cocktail effect," in HEAs was first proposed by Ranganathan in 2003[13]. Although the harmonizing properties of metals also exist in traditional metals and metallic glasses, in the field of HEAs, there are more elements and phases that can contribute, and the phenomena are more diverse. The nanostructures and phase properties should simultaneously contribute to the overall alloy properties. In 2006, Yeh et al.[14] demonstrated the Al$_x$CoCrCuFeNi alloy system, which showed that by increasing the aluminum content, the interaction between elements can transform the alloy from face-centered-cubic (FCC) to BCC phase. The other classic example used to demonstrate the cocktail effect is in refractory HEAs[15]. For example, the MoNbTaW and MoNbTaVW have melting points higher than 1650 °C for each component, and the melting point of the alloy after mixing is higher than 2600 °C. The softening temperature of this alloy is better than that of superalloys, and the yield strength is higher than 400 MPa at 1600 °C.

In this work, we present a research methodology for the HEA-conductivity model and further develop the material composition of Nb-Mo-Ta-W to make it both conductive and wear-resistant that can be applied to atomic force probe coating. The cocktail effect on the conductivity of HEA films, defined as the variation of conductivity achieved through linear combination of constituent elements, is exploited and examined. Both simulations and experiments have been used to discuss how compositional modulation affects conductivity in HEAs.

## Results and discussion
### Calculation of a conduction model
To build a conduction model of HEAs, we start from the fundamental theory of free electrons in a conductor. According to a classical-electromagnetic theory, the frequency-dependent conductivity can be represented by the Drude-Lorentz model[16]. The Drude model considers the conductivity contributed by the displacement of free electrons, i.e., intraband contributions[17]. The Lorentz model takes into account the conductivity contributed by electrons bound by atomic nuclei, i.e., interband contributions[18]. Therefore, the conductivity behavior under a low-frequency external electric field is mainly described by the Drude model, while the conductivity behavior under a high-frequency (greater than $10^{12}$ Hz) external electric field is primarily described by the Lorentz model.

Calculations were conducted to compare three evaluation methods in an attempt to find a more feasible approach to predict the low-frequency conductivity of HEAs. The three compared methods are described as follows: (1) Method 1: This is the traditional linear combination method. It starts with the conductivity of pure elements and estimates the conductivity of HEAs composed of these elements by linearly combining their compositions. This method does not consider the plasma frequency due to random changes in coordinating elements, nor does it consider the impact of lattice random distortions on relaxation time. (2) Method 2: This technique considers the plasma frequency due to random changes in coordinating elements but does not consider the impact of lattice random distortions on relaxation time. For example, the plasma frequency is calculated from a solid-solution model of HEAs using first principles, while the relaxation time is calculated employing a linear combination of composition components and then utilized in the Drude model to calculate the low-frequency conductivity. (3) Method 3: This scheme considers both the plasma frequency due to random changes in coordinating elements and the impact of lattice random distortions on relaxation time. For instance, the plasma frequency is calculated from a solid-solution model of HEAs using first principles, while the relaxation time is extracted from measurements using terahertz time-domain spectroscopy (THz-TDS), which is an advanced technique for detecting changes in the electric-field amplitude and phase as it passes through a sample, allowing for the calculation of material dielectric properties and complex conductivity characteristics[19–22]. Both values are then used in the Drude model to calculate low-frequency conductivity. The detailed calculation method is given in Supplementary Note 1, and the measurement of the relaxation time is included in Supplementary Note 3 and Supplementary Figs. 4, 5.

By comparing these three methods, the simulation work aims to identify the most suitable approach for predicting the low-frequency conductivity of HEAs. The results obtained from these three methods will be compared with the experimentally measured DC conductivity. It is anticipated that the results from Method 3 will be the most accurate. Comparing the results of these three methods will provide insights into the impact of random changes in coordinating elements on plasma frequency and the effect of lattice structure random distortions on relaxation time in HEAs. This comparison will also help understand the extent to which these factors alter low-frequency conductivity relative to the linear combination of pure elemental properties.

The calculation results of the frequency-dependent conductivity, plasma frequency, and free-electron density are given in Supplementary Note 2. Comparing the effective-electron masses among the three HEAs, it is observed that the equimolar model has the largest effective-electron mass; the Ta plus W comprising 70 mol% model has the second-largest effective electron mass, which is close to that of the equimolar model; the Mo plus W comprising 70 mol% model has the smallest effective electron mass. This trend in the effective electron mass is similar to that observed in plasma frequencies. However, it is important to note that the order of the equimolar model and the Ta plus W comprising 70 mol% model is reversed when compared to the plasma-frequency trend. This feature suggests that the formation of HEAs significantly affects the effective-electron mass, with variations based on the specific alloy composition. Therefore, when multiple elements form HEAs, the free-electron plasma frequency significantly decreases and deviates from linear-combination values based on the elemental composition, with a reduction of approximately half of the

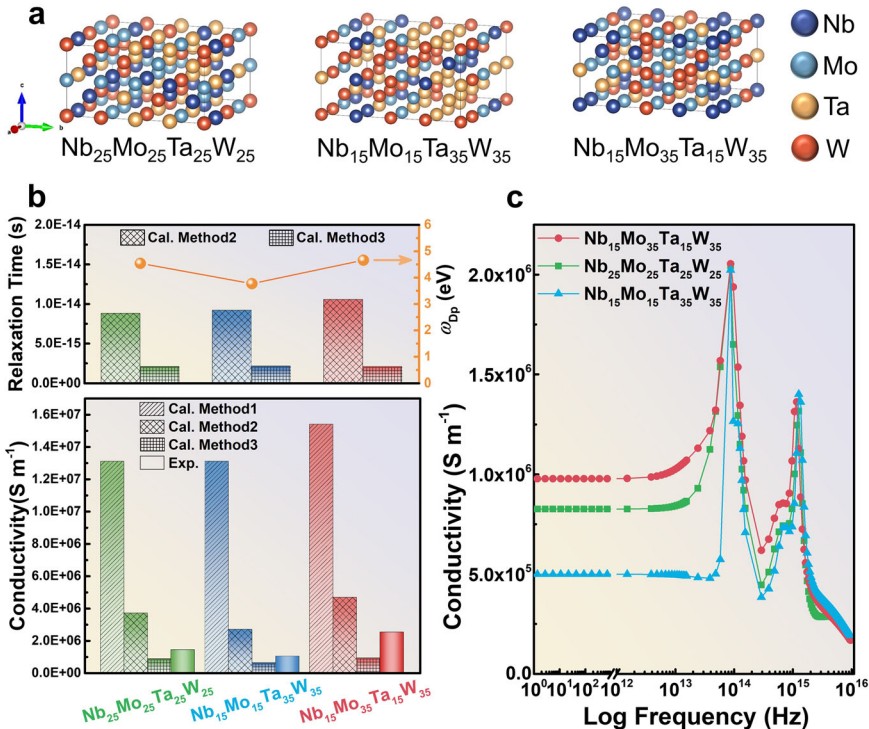

**Fig. 1 | Atomic model and comparative studies of conductivity from different methods and experiment result.** Method 1 is the conventional linear combination approach, which estimates the conductivity of high-entropy alloys by linearly combining the conductivity of pure elements. Method 2 considers the plasma frequency variations due to the random distribution of coordinating elements but does not account for the influence of lattice random distortion on relaxation time. The plasma frequency is calculated from first principles using a solid-solution model for high-entropy alloys, while relaxation time is obtained through a linear combination of the compositional components and calculated using the Drude model for low-frequency conductivity. Method 3 also considers the plasma frequency variations due to the random distribution of coordinating elements and accounts for the influence of lattice random distortion on relaxation time. The plasma frequency is calculated from first principles, while relaxation time is extracted from measurements using Terahertz (THz) time-domain spectroscopy (THz-TDS) and subsequently used in the Drude model for low-frequency conductivity calculations. **a** Atomic model derived from computation. **b** Comparing the DC conductivity by predictions using three methods. **c** Frequency dependent conductivity using Drude and Lorentz models (See more calculation details in Supplementary Note 1 to Note 3).

linear combination value. This trend is primarily due to a substantial decrease in the free-electron density and an increase in the effective-electron mass. Consequently, it is not possible to predict the plasma frequency of HEAs through a linear combination of the plasma frequencies of individual elements.

Subsequently, the Drude and Lorentz models were used to fit the dielectric constant obtained using Method 1, as presented in Eq. 1.

$$\varepsilon_{DL}(\omega) = \varepsilon_\infty - \sum \frac{\omega_{Dp}^2}{\omega^2 + i\omega\gamma_d} - \sum \frac{\omega_{Lp}^2}{\omega^2 - i\omega\gamma_L - \omega_L^2} \quad (1)$$

Here, $\varepsilon_\infty$ represents the dielectric constant at infinite frequency, which is ideally 1 for an electron gas. $\omega_{Dp}$ and $\omega_{Lp}$ are the plasma frequencies for the Drude and Lorentz models, respectively. $\gamma_d$ is the electron-collision frequency, $\gamma_L$ is the damping coefficient, and $\omega_L$ is the resonance frequency. By substituting the fitted parameters into Eq. 2, the frequency-dependent conductivity is obtained.

$$\sigma_{DL}(\omega) = \sum \frac{\varepsilon_0 \omega_{Dp}^2 \tau_D}{1 - i\omega\tau_d} - \sum \frac{i\varepsilon_0 \omega_{Lp}^2 \omega}{-\omega^2 - i\omega\gamma_L - \omega_L^2} \quad (2)$$

### Conductivity of Nb-Mo-Ta-W HEA films

Figures 1a–c show the atomic models, relaxation times, plasmon frequency, and conductivities of the three different Nb-Mo-Ta-W films with different estimation methods. Compared with Fig. 2a, which shows the experimental results of resistivity as a function of thickness,

it is obtained that when the Mo and W components are increased to 35%, the resistivity of the Nb$_{15}$Mo$_{35}$Ta$_{15}$W$_{35}$ film decreases, compared to the equimolar Nb$_{25}$Mo$_{25}$Ta$_{25}$W$_{25}$. Conversely, when Ta and W are increased to 35%, the resistivity of the Nb$_{15}$Mo$_{15}$Ta$_{35}$W$_{35}$ increases. This trend is consistent with calculation results in each thickness. It reveals that the resistivity of HEAs can be modulated by the inherent conductivity of the elements. Method 1 yields conductivity values that are approximately one order of magnitude higher than the experiment, indicating that applying the ideal solution concept, i.e., cocktail effect, to estimate conductivity in solid-solution HEAs can only provide a qualitative assessment. Method 2, on the other hand, aligns more closely with experimental values and falls within the same order of magnitude, demonstrating that considering the plasma frequency when forming HEAs can significantly improve predictive accuracy. As for Method 3, the calculated conductivities are even closer to experimental values, with errors of within 50%, highlighting that considering the impact of relaxation time can further enhance predictive accuracy. Regardless of the method used, all of them accurately predict the trends in composition changes, matching experimental values.

To further discuss the relationship between the linear resistivity and the measured resistivity of different material systems, as demonstrated by the three material systems in Supplementary Fig. 6. The first system is the Al-Zn-Co-Ni alloy recently published by our team[23], the second is the Co-Cr-Fe-Ni-Nb alloy[24], and the third is the Nb-Mo-Ta-W alloy proposed in this study. In all three high-entropy alloy series considered, a robust linear correlation is evident. This finding underscores a clear linear association between measured resistivity and

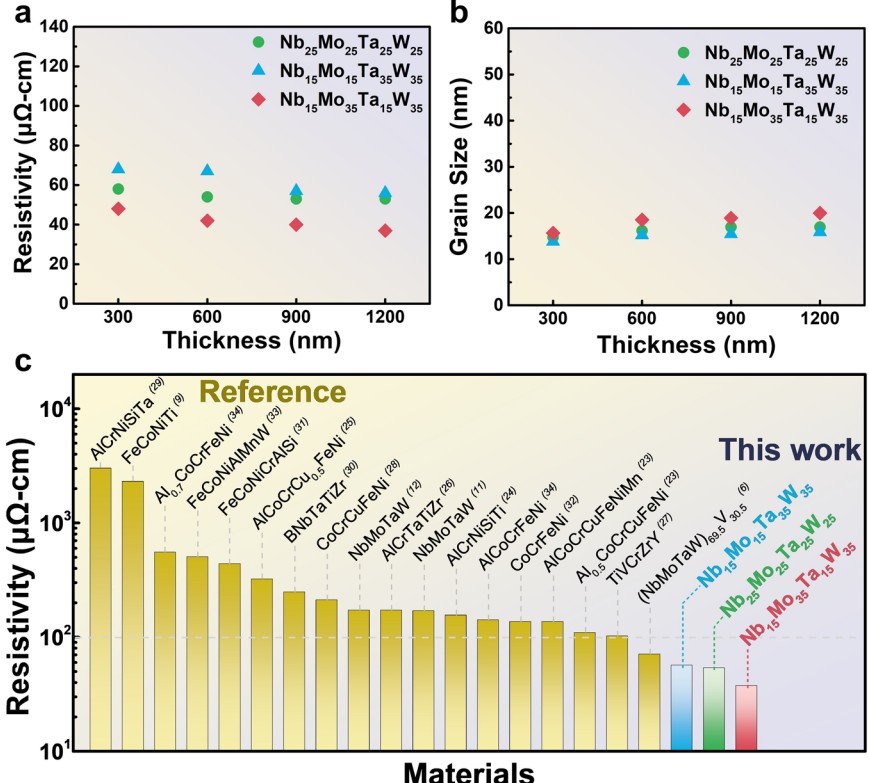

**Fig. 2 | Electrical resistivity with thickness and grain size of NbMoTaW films. a** Resistivities versus thickness. **b** Grain size versus thickness. **c** Comparison of the resistivities of $Nb_{25}Mo_{25}Ta_{25}W_{25}$, $Nb_{15}Mo_{15}Ta_{35}W_{35}$, and $Nb_{15}Mo_{35}Ta_{15}W_{35}$ films with those of other high entropy alloy films in the literature[6,9,11,12,26–37].

composition-weighted resistivity in HEA. This also reaffirms that the cocktail effect can be utilized to predict trends in HEA conductivity, albeit with orders of magnitude differences. To gain deeper insights into the conduction mechanisms of various HEAs, the computational methods proposed in this study can be employed.

**Grain-size and phase-composition effects on HEA conductivity**

The variation in the resistivity for Nb-Mo-Ta-W films with different compositions at various thicknesses can be observed. The resistivity is the smallest when Mo plus W comprises 70 mol%, the largest when Ta plus W comprises 70 mol%, and intermediate for the equimolar composition (Fig. 2a). The analysis of the X-ray diffraction (XRD) full width at half maximum can determine the grain size using the Scherrer formula, $D = \beta \cos\theta / K\lambda$, where $D$ is the grain size, $K$ is the shape factor, $\lambda$ is the wavelength of the X-ray, $\beta$ is the full width at half maximum of the peak, and $\theta$ is the Bragg angle[25]. The differences in grain size due to variations in composition are around 15–17 nm (Fig. 2b), which should have no impact on the electrical resistivity. Compared to the resistivity reported in the literature[6,9,11,12,26–37], which is mostly size independent. Moreover, although the XRD calculations yielded a grain size of only a few tens of nanometers, the NbMoTaW thin film exhibits high conductivity, which is attributed to its crystalline structure. The observed lamellar shape in SEM (Supplementary Fig. 7) has a longitudinal length of several hundred nanometers, suggesting it serves as the primary conduction direction. Therefore, the synthesized solid-solution thin films in this study can initially eliminate the influence of nanostructures, grain size, and precipitation on electrical conductivity, focusing specifically on the effects of lattice distortion and compositional modulation. When the thickness approaches 1200 nm, the resistivity of $Nb_{15}Mo_{35}Ta_{15}W_{35}$ can be lowered to ~37 μΩ·cm. Among other high-entropy films, the lowest resistivity observed in the literature is for the AlCoCrFeNi and CoCrFeNiZr systems, which are close to 100 μΩ·cm (Fig. 2c). The $Nb_{15}Mo_{35}Ta_{15}W_{35}$ films synthesized herein

have a room-temperature resistivity lower than 100 μΩ·cm and exhibit superior mechanical properties (to be discussed below), creating possibilities for novel applications of these high-entropy films. Notably, when the thickness is increased to 2 μm, the change in resistivity is not significant, indicating that the influence of thickness on resistivity in our sample is minimal.

The XRD diffraction patterns of $Nb_{25}Mo_{25}Ta_{25}W_{25}$, $Nb_{15}Mo_{35}Ta_{15}W_{35}$, and $Nb_{15}Mo_{15}Ta_{35}W_{35}$ thin films (Supplementary Fig. 8a, b) can be clearly seen that all three films have a BCC structure, and the (110) diffraction peaks are located between the low-angle peaks of Ta and W and the high-angle peaks of Mo and W. Increasing the W and Ta to 35% shifts the diffraction peaks towards lower angles slightly, while increasing the Mo and W to 35% shifts them towards higher angles markedly. Adjusting the ratio of Mo, W, and Ta does not change the BCC solid-solution phase. Supplementary Fig. 9 presents that the calculated XRD patterns based on the MD simulation fit well with the experiment. The lattice constants calculated from the diffraction peaks decrease with increasing the Mo and W ratio but decrease with increasing the Ta and W ratio (Supplementary Table 6 and Supplementary Fig. 10). This trend is in complete agreement with the simulation results, as summarized in Supplementary Table 5. High-resolution TEM selected area diffraction (Supplementary Fig. 8c, d) shows diffraction rings that match the BCC structure. The lattice phase obtained by the Fourier analysis of ten crystal planes with an average lattice spacing of ~2.313 Å corresponds to the (110) plane, which is close to the value observed by XRD. The elemental distribution of the $Nb_{25}Mo_{25}Ta_{25}W_{25}$ thin film obtained by TEM (Supplementary Fig. 8e), and it can be seen that the composition of the four elements is evenly distributed. Simultaneously, SEM was also used to further evaluate the distribution of elemental composition over a larger area, as shown in Supplementary Fig. 11. At the micron scale, the elemental composition distribution in all films is very uniform, with no apparent segregation observed. Therefore, the influence of phase composition inhomogeneity on conductivity can be

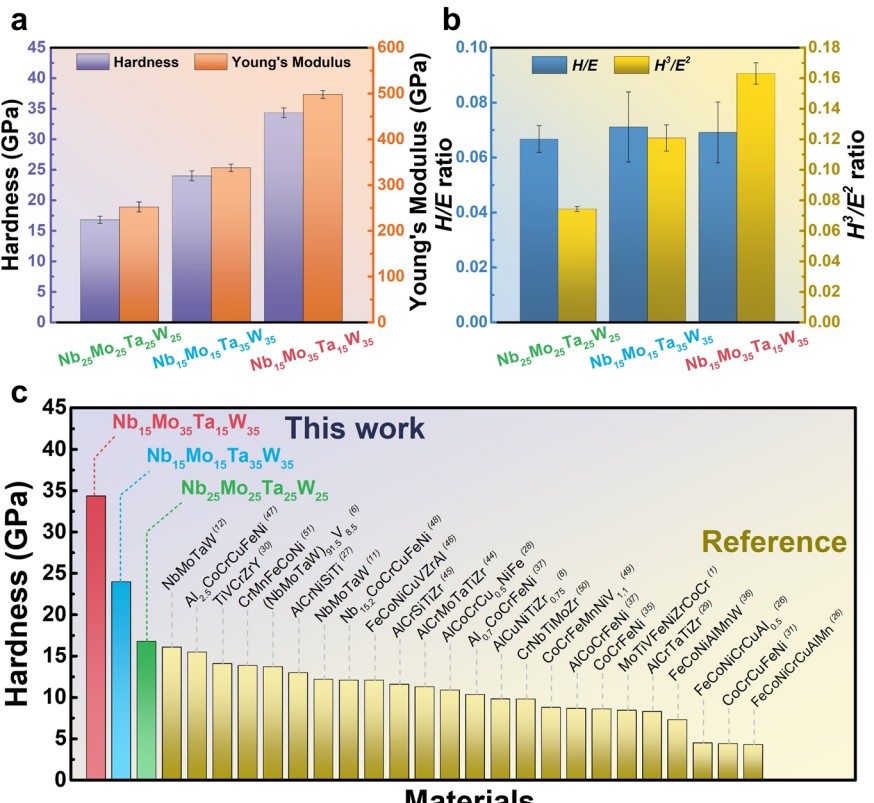

**Fig. 3 | Mechanical properties of NbMoTaW films. a** Hardness and Young's modulus. **b** The $H/E$ and $H^3/E^2$ ratio. The error bars represent the standard deviation. **c** Comparison of the hardness of $Nb_{25}Mo_{25}Ta_{25}W_{25}$, $Nb_{15}Mo_{15}Ta_{35}W_{35}$, and $Nb_{15}Mo_{35}Ta_{15}W_{35}$ films with those of other high entropy alloy films in the literature[1,6,8,11,12,26–31,35–37,44–51].

initially ruled out in our experiments. Short-range order (SRO) characterizes the local atomic structure and can be distinct from the long-range order, which describes the overall periodicity or arrangement of atoms throughout the crystal lattice. Recently, many studies have indicated the impact between mechanically derived short-range order and multi-principal-element[38–41]. Although no second phase is found by XRD and TEM, THz TDS is also used to clarify the existence of SRO that should influence on the electrical and mechanical properties of HEAs. Because of the interaction with lattice vibrations and electronic transitions within the THz frequency range, THz TDS can differentiate between different phases or alloys within a metal sample based on their unique frequency-dependent THz conductivity. In other words, the periodicity and magnitude of the oscillations in THz conductivity can carry information about the local structural features associated with short-range order. By studying the behavior of complex conductivities in our HEAs nano thin film, there is no any feature of oscillation. Therefore, it is apparently not sensitive to the local atomic arrangements and bonding configurations within the short-range ordering of a crystalline material.

### Surface morphology and conductivity using atomic-force microscopy (AFM)

To understand the relations among the composition, surface morphology, and surface conductivity, conductive AFM measurements were conducted, and the surface morphology and electric current images were also shown in Supplementary Fig. 12. The surface-current distributions are dependent on the alloy compositions and the flake morphology. In the current mapping images, the white area indicates high surface current measurements. The high-current coverage percentages for $Nb_{25}Mo_{25}Ta_{25}W_{25}$, $Nb_{15}Mo_{15}Ta_{35}W_{35}$, and $Nb_{15}Mo_{35}Ta_{15}W_{35}$ are 96.13%, 95.49%, and 98.89%, respectively. Since

the films are of the same thickness of 1200 nm, the flakes seem connected to form a conductive network, the conductive current mapping covered most area. Furthermore, the $Nb_{15}Mo_{35}Ta_{15}W_{35}$ composition has the best surface-film formation. The effective current of C-AFM images was calculated by integrating the total current pixel-by-pixel, for $Nb_{25}Mo_{25}Ta_{25}W_{25}$, $Nb_{15}Mo_{15}Ta_{35}W_{35}$, and $Nb_{15}Mo_{35}Ta_{15}W_{35}$, the values are 21.52 μA, 18.32 μA, and 25.39 μA, respectively (Supplementary Fig. 13 and Supplementary Table 7). If all three films had a 100% surface-current coverage, the effective current for them would be 22.29 μA, 19.19 μA, and 25.67 μA. Since the conductivity of an alloy is determined mainly by its composition, we concluded that the conductivity of these three films is: $Nb_{15}Mo_{35}Ta_{15}W_{35}$ > $Nb_{25}Mo_{25}Ta_{25}W_{25}$ > $Nb_{15}Mo_{15}Ta_{35}W_{35}$, agree-well with the calculation and experimental results stated above.

### Mechanical properties

According to the literature[42,43], both hardness ($H$) and Young's modulus ($E$) are related to the wear resistance of thin films. While high hardness is directly associated with great wear resistance, a high elastic modulus can lead to stress concentration, thereby reducing the wear resistance of films. Elasticity indicators can be defined by $H/E$, and plasticity indicators like ($H^3/E^2$) are commonly used to assess the ability to resist wear of films because they are positively correlated with toughness. To compare the wear-resistance characteristics of the three thin films, the films were deposited individually onto silicon substrates with a thickness of 1000 nm. The results in Fig. 3a revealed that the $H/E$ values for all three components were similar, indicating similar abilities during the elastic-deformation phase. However, the film with Mo and W as dominant components (35%) exhibits the highest hardness and simultaneously has the greatest $H^3/E^2$ value, as shown in Fig. 3b signifying the highest wear resistance capability. The continuous method

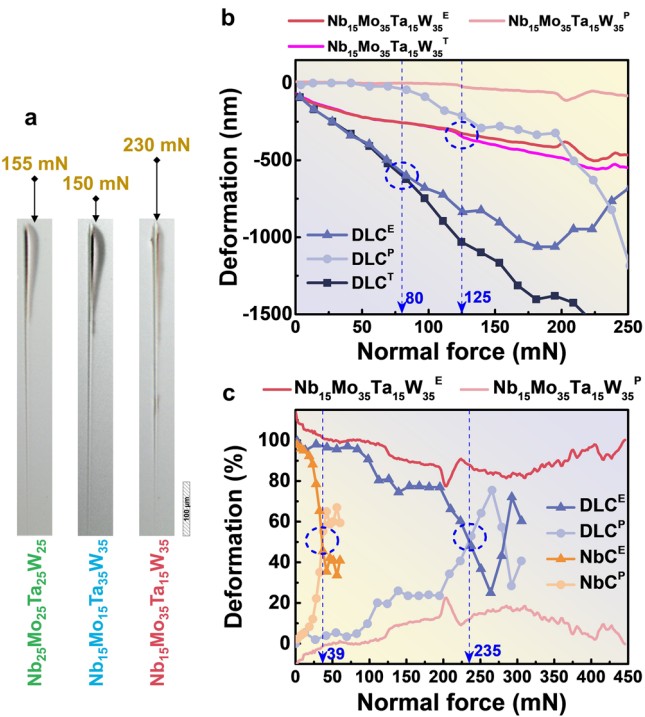

**Fig. 4 | Adhesion characteristics analyzed by nanoscratch testing.**
**a** Nanoscratch test results of NbMoTaW thin films. **b** Variation in elastic-plastic deformation contribution with depth for high-entropy films and diamond-like carbon films. **c** Proportional change in elastic-plastic deformation for high-entropy films, diamond-like carbon films, and wear-resistant NbC ceramic films with applied load. E stands for Elastic, P stands for Plastic, T stands for Total.

employed to measure the relationship among the stiffness, hardness, and Young's modulus are shown in Supplementary Figs. 14 to 16, respectively. Figure 3c demonstrates that compared to the HEA films reported in the literature[1,6,8,11,12,26–31,35–37,44–51], the NbMoTaW films prepared in this study exhibit the highest level of hardness.

In nanoscratch tests, plastic deformation increases both the energy absorption and friction force while the Berkovich tip scratches the sample surface. From the optical-microscope images of the scratch tests, the scratched surface of HEA films did not show signs of delamination or fractures. Hence mainly elastic and plastic deformation of films are considered. As indicated in Fig. 4a, the onset of abrasion and the occurrence of pileups occur at 155 mN, 150 mN, and 230 mN for the equal-molar, Ta and W dominate (both 35%) and Mo and W dominate (both 35%) films, respectively, agreeing with the trend of $H^3/E^2$ trend. Figure 4b shows the deformation depth versus normal force applied on the NbMoTaW film during nanoscratch tests. Also plotted are the results of referenced diamond-like-carbon (DLC) films[52]. The HEA film started to yield and had plastic deformation at a normal force of 125 mN, while DLC films yielded at 80 mN. The deformation process became dominated by plastic deformation when the total deformation is above 50%, as indicated by a critical load in Fig. 4c. At the maximum normal force of 450 mN during the nanoscratch tests, the HEA films did not reach critical loading and have overall less than 20% plastic deformation, while the referenced DLC and NbC films reached critical loading at 235 mN and 39 mN, respectively[53]. Detailed data of the nanoscratch test is given in Supplementary Fig. 17.

To simultaneously investigate the environmental weatherability of HEA films, potentiodynamic polarization curves were measured using electrochemical methods to analyze the corrosion behavior. The results showed that the corrosion currents ($i_{corr}$) of HEA films with thicknesses of 300 nm and 1200 nm were very low under both conditions of 3.5 wt.% NaCl solution (Supplementary Fig. 18a) and 0.1 M

$H_2SO_4$ solution (Supplementary Fig. 18b), approximately ~$8 \times 10^{-7}$ A and ~$2 \times 10^{-6}$ A, respectively, and were independent of the thickness. These characteristic values, as listed in Supplementary Table 8, under the common Standard ASTM G44 conditions, indicate that the corrosion resistance of HEA films is superior to 304 steel[54] and γ-TiAl[55], and closely comparable to SS316L stainless steel[56], demonstrating good corrosion resistance and environmental weatherability.

The results of the calculations and experimental findings indicate a positive correlation between the composition-weighted properties of HEAs and those obtained through experiments or first-principles calculations. Supplementary Note 4 summaries these results. The Supplementary Table 9 and Fig. 19 demonstrates that all considered mechanical properties conform to the trendline, indicating the feasibility of estimating the mechanical characteristics of HEAs through a linear combination of their constituents. Additionally, the transformation temperatures from the HEA phase to the B2 phase are plotted in Supplementary Fig. 20 based on the composition ratios of Nb, Mo, and W. It suggests that if the composition ratio of Mo and W exceeds 35 mol%, there needs to be at least 0.15 mol% of Nb present to ensure that the HEA phase transformation temperature remains below 400 °C. Therefore, considering both the phase transition temperature and the mechanical properties and conductivity, the composition ratio of Nb$_{15}$Mo$_{35}$Ta$_{15}$W$_{35}$ seems to strike a close-to-optimal balance.

## Conductive and wear performance of the NbMoTaW coatings on AFM probe tips

Except for the bare silicon tip, which is not conductive, all of the HEA-coated tip remained high-conductive throughout the wear tests (verified by the conductive atomic-force microscope (C-AFM) scan images, not shown). The evolution of the tip apex during the in-use conductive wear tests, as estimated by a blind-tip reconstruction method is available in Fig. 5a. The detailed wearing test method is given in Supplementary Fig. 21. To maintain the high-resolution scan capability of the AFM tip, the HEA coatings have to be minimal, a typical conductive coating thickness for a commercial probe is from a few nm to about 20 nm. Figure 5b suggested that both compositions of Nb-Mo-Ta-W coatings already exhibited high in-use wear resistance under simultaneous mechanical loading and electrical bias - about a 70–80% reduction of the tip-wear volume, as compared with bare silicon tips. The reduction in wear for HEA coatings provides potential for high-conductive low-wear applications. Figure 5c denotes the lowest applied bias to achieve stable electrical measurements with respect to the HEA-coating thickness of the AFM probes. Comparing compositions, Nb$_{25}$Mo$_{25}$Ta$_{25}$W$_{25}$ and Nb$_{15}$Mo$_{35}$Ta$_{15}$W$_{35,}$ the latter exhibits better performance since the required bias is at least 50% lower than that of the former for all three coating thicknesses tested. This result coincided with the effective surface-current measurement in Supplementary Fig. 10. Figure 5d compares the in-use wear performance of a commercial conductive (Pt/Ir coated) AFM probe with that of a Nb$_{15}$Mo$_{35}$Ta$_{15}$W$_{35,}$ coated probe. During the conductive current mapping, a constant force of 12 nN and electrical voltage bias of 500 mV were applied between the AFM tip and the sample surface. The commercial tip showed a stable surface-current signal at the beginning of the test and became almost no current signal after it had traveled on the sample surface for 10.75 mm in distance. On the other hand, the Nb$_{15}$Mo$_{35}$Ta$_{15}$W$_{35}$coated probe remains a stable current signal during force-asserted C-AFM imaging even after the tip has traveled for 19.71 mm. Compared with conventional conductive metals like Cu, Ag, Al, and Au films, HEA films are much hard. Compared with the hard coating films like TiN, TaN, and NbC, the proposed Nb-Mo-Ta-W films are also harder and more conductive. The relevant comparative illustrations can be found in Fig. 5e[6,11,12,26–31,35–37,57–61].

Finally, Nb$_{15}$Mo$_{35}$Ta$_{15}$W$_{35}$ thin films were deposited onto the ReW probe used for semiconductor testing and placed it on a probe card for wear testing (Supplementary Fig. 22). Supplementary Fig. 23 shows the

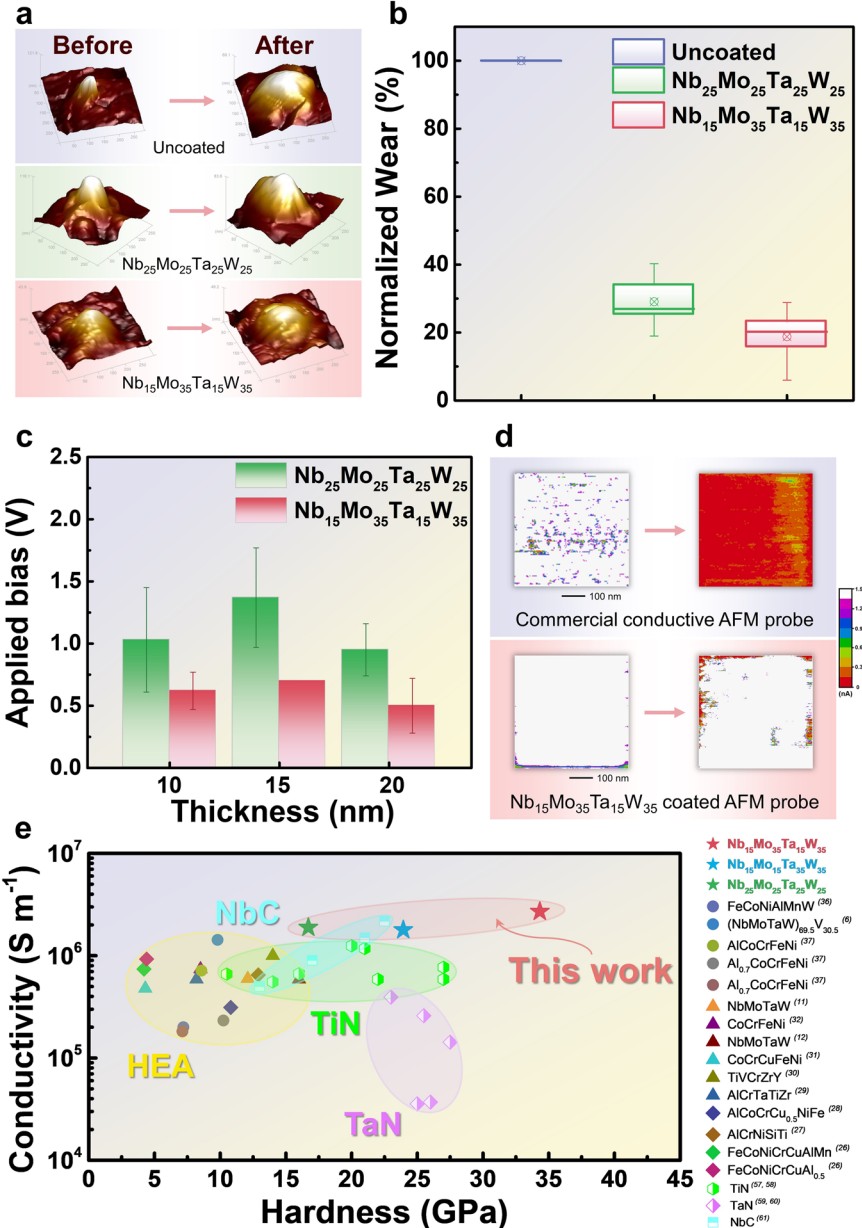

**Fig. 5 | Conductive and wear performance of the NbMoTaW coatings on AFM probe tips. a** Uncoated (bare silicon tip), $Nb_{25}Mo_{25}Ta_{25}W_{25}$, $Nb_{15}Mo_{35}Ta_{15}W_{35}$ changes in surface morphology before and after wear. **b** In-use wear resistance under simultaneous mechanical loading and electrical bias. **c** Required bias to achieve the same current level for different thickness. $Nb_{15}Mo_{35}Ta_{15}W_{35}$ shows lower bias than $Nb_{25}Mo_{25}Ta_{25}W_{25}$, indicating the former has the better surface conductivity. The error bars represent the standard deviation. **d** In-use wear performance of a commercial conductive (Pt/Ir coated) AFM probe with that of a $Nb_{15}Mo_{35}Ta_{15}W_{35}$ coated probe. **e** Comparison of the hardness and conductivity of $Nb_{25}Mo_{25}Ta_{25}W_{25}$, $Nb_{15}Mo_{15}Ta_{35}W_{35}$, and $Nb_{15}Mo_{35}Ta_{15}W_{35}$ films with those of other high entropy alloy films and hard coating films in the literature[6,11,26–31,35–37,57–61]. Note that only thin film samples are considered.

results of a wear test with 200,000 probing cycles, indicating a significant improvement in the wear resistance of the ReW probe, demonstrating a high potential for industrial applications. In addition to probe coating, such highly conductive and hard HEA thin films can be applied to connectors and interfaces in electronic products, high-voltage switches in power systems, starters in automobiles, and electrodes in precision measuring instruments, ensuring long-term durability and reliable electrical connections.

In summary, this study has established a conductivity model for the HEA system Nb-Mo-Ta-W, designed alloy compositions with high conductivity and wear resistance, which can be applied to enhance the atomic-level resolution and lifespan of atomic force microscope probes. By adjusting the composition, a refractory high-entropy material of $Nb_{15}Mo_{35}Ta_{15}W_{35}$ was endowed with both high hardness

and electrical conductivity. We indicate that the conductivity of Nb-Mo-Ta-W HEAs can be modulated through the cocktail effect but tends to differ by an order of magnitude compared to linear combinations. The reasons for the decrease in conductivity are attributed to changes in the plasma frequency and relaxation time resulting from lattice distortions, alterations in the effective mass and density of free electrons, and variations in the crystalline size after the formation of HEAs.

## Methods

### Thin film preparation and characterization

The P-type Silicon (100) single-sided polished substrates with high resistivity (~ 10 kΩ·cm) were pre-treated by a standard RCA (Radio Corporation of America) cleaning. The material of target, respectively, used $Nb_{25}Mo_{25}Ta_{25}W_{25}$, $Nb_{15}Mo_{15}Ta_{35}W_{35}$, and $Nb_{15}Mo_{35}Ta_{15}W_{35}$ alloy

targets with an equal proportion of elemental compositions and 99.99 at.% purity. Before starting the deposition, the base pressure was below $5 \times 10^{-6}$ Torr. After pre-sputtering for 10 min, the $Nb_{25}Mo_{25}Ta_{25}W_{25}$, $Nb_{15}Mo_{15}Ta_{35}W_{35}$, and $Nb_{15}Mo_{35}Ta_{15}W_{35}$ thin films with 150 nm to 1200 nm were then deposited by DC-magnetron sputtering, respectively, by setting the working pressure of the deposition was to 1.5 mTorr with a 40 sccm Ar flow.

The crystalline structure was examined by grazing the incidence X-ray diffraction (GIXRD, Bruker D8 DISCOVER) with Cu-Kα ($\lambda = 0.15406$ nm) radiation. The diffraction angle (2θ) scanned from 20° to 100°. The surface morphology and film thickness were analyzed, using high-resolution scanning electron microscopy (HR-SEM, HITACHI SU8000). An investigation of the elemental composition was conducted, employing an X-ray energy dispersive spectrometer (EDS, Bruker XFlash 5060 F) linked with HR-SEM. The transmission electron microscope (TEM, JEOL JEM-2010F) was used to observe the detailed microstructure and confirm the phase structure. The resistivity of thin films was measured with the size of $2 \text{ cm} \times 2 \text{ cm}$, using a four-point probe measurement system with a Keithley 2400 Source Meter.

The mechanical properties of thin films were evaluated by the nanoindentation systems (Nanoindenter, MTS XP and MTS G200). In a nanoindenter system, a Berkovich probe made of a three-faced single crystal diamond applied loading to the specimen surface. Based on the Oliver-Pharr method[62], the mechanical properties, such as elastic modulus, can be accessed from the load-displacement curves. The indenter system used a continuous stiffness measurement (CSM) mode, which applies a dynamic cyclic loading during the indentation of a diamond tip into the sample surface. For the HEA films, CSM has the advantages of probing the local mechanical responses with high sensitivity on materials compositions.

The nanoscratch tests were conducted with the indenter system where the diamond tip applied increasing normal loading while indenting on the sample surface. The friction coefficient can be calculated, and the forms of material pileup debris also provide information of the materials' wear behaviors. The test consists of three steps: (1) Pre-profiling, the tip traces the sample surface with only 50-μN loading to acquire the information of the surface profile. (2) Scratching (ramp-load), the tip scratches the sample surface with increasing the loading force to 450 mN while the penetration curve is recorded for the calculation of friction forces and coefficients. (3) Post-profiling, the tip traces the sample force again with 50 μN loading to acquire the information of the surface profile after the scratch.

## Probe coating and testing

One of the potential applications of our HEA coatings is for high-conductive, high-wear-resistive overcoat for electrical-measurement probes. In this application, both the conductivity and wear resistance of the HEA coatings must be evaluated simultaneously. Hence, we designed an in-use conductive wear test by depositing HEA thin films onto AFM probe tips, and then use the as-deposited AFM tips to scan a high conductive Cu sample in the conductive AFM (C-AFM) scan mode. While scanning the surface, an electrical bias of + 1.5 V was applied to the sample to show the wear characteristics at high currents. The C-AFM scan total distance is 10.24 mm with a consistent applied force of 61 nN.

Tip wear can be characterized, using the "TipCheck" (Aurora NanoDevices Inc., Canada) standard characterizer sample to compare the difference of the tip radius as well as the shape before and after the wear test. The "blind tip reconstruction" algorithm was adopted to estimate the effective tip radius of the tip apex from the TipCheck scan data[63-65].

## Computational methods

In this study, the first-principles calculation package, VASP, based on the DFT theory developed by the Institut für Theoretische Physik of Technische Universität Wien was used[66]. The interactions between the ionic cores and valence electrons were modeled, using the projector-augmented wave (PAW) potential[67]. The generalized gradient approximation (GGA) with the parameterization of Perdew-Burke-Ernzerhof (PBE) was used as the exchange-correlation functional in this study[68]. The valence electrons of the elements considered, including $Nb\_pv(4p^6 4d^4 5s^1)$, $Mo\_pv(4p^6 4d^5 5s^1)$, $Ta(5d^3 6s^2)$, and $W(5d^4 6s^2)$, were included in the calculations. Spin-polarized calculations based on the theory proposed by Rudolf Zeller were also considered[69]. A plane wave cut-off energy was expanded until a kinetic energy convergence of 500 eV was achieved. The models were subjected to structural optimization on a $3 \times 3 \times 3$ k-point mesh that was tested by the convergence test. The ionic relaxation converges were set as $10^{-8}$ eV. The criterion for geometry optimization was defined as the action of the Hellman-Feynman forces on atoms that were less than 0.02 eV/Å. The optical properties for the materials in this study are calculated, using the independent-particle approximation (IPA) method.

We built the atomic model for the HEA, featuring a disordered arrangement of atoms, using the inverse Monte Carlo algorithm[70]. This algorithm generates a model where each type of atom adheres to a coordination number matrix, specifying the likelihood of being the nth nearest neighbor for all pairs of atom types. Our atomic models for NbMoTaW consisted of $3 \times 3 \times 3$-unit cells with a BCC structure. We created three models for each composition and calculated the average values to derive the material properties.

## Data availability

The data supporting the findings of this study are interpreted, verified, and presented in main text and the Supplementary Information, and are available from the corresponding authors upon request.

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

## Acknowledgements
This work was financially supported by the National Science and Technology Council in Taiwan under Grant Nos. NSTC 112-2224-E-006-004 (C.F.S.) and 112-2221-E-006-076 (C.F.S.), and by the Taiwan Semiconductor Research Institute under JDP113-Y1-028 (C.F.S.). The authors greatly appreciate the ongoing financial and technical support provided by the Applied High Entropy Technology (AHET) Center. This research was supported in part by a Higher Education Sprout Project, Ministry of Education to the Headquarters of University Advancement at National Cheng Kung University. It also received financial support from the Hierarchical Green-Energy Materials (Hi-GEM) Research Center at National Cheng Kung University, which is part of the Featured Areas Research Center Program within the framework of the Higher Education Sprout Project by the Ministry of Education in Taiwan. The authors appreciate the support from the National Science Foundation (DMR-1611180, 1809640, 2226508) (P.K.L.) and the Army Office Project (W911NF-13-1-0438 and W911NF-19-2-0049) (P.K.L.). The authors appreciate the HR-SEM (Hitachi SU8000) and EDS (Bruker XFlash 5060 F) belonging to the Core Facility Center of National Cheng Kung University.

## Author contributions
C.H.Y. contributed to the sample preparation, thin-film coating, characterization of materials, and writing. W.D.H., C.Y.K., Y.C.C., and K.S.H. contributed to the writing and simulation works, including conductivity, plasma frequency, lattice distortion, effective mass, free electron density, and lattice constant. B.H.L. and S.S.J. contributed to the AFM measurements, C-AFM analysis, nanoscratch, hardness measurements, and writing. C.S.Y. and C.Y.L. contributed to the terahertz time-domain spectroscopy measurement for relaxation time and writing. P.K.L. contributed to the investigation of mechanical properties, and assistance with article refinement. C.F.S. contributed to research design, writing, project organization, and execution. All authors contributed to discussions and writing.

## Competing interests
The authors declare no competing interests.
