## [Peer Review File · Nature Communications]

Low-frequency conductivity of low wear high-entropy alloysREVIEWER COMMENTS

Reviewer #1 (Remarks to the Author):

This paper integrates theoretical model, simulation results and experimental data to understand the impact of "cocktail effect" and lattice distortions on electrical conductivity of HEAs. The findings suggest that the Nb-Mo-Ta-W alloy exhibits both high hardness and good conductivity, making it suitable for atomic force microscopy probe coating, significantly improving their wear resistance and atomic-scale image resolution. Overall, this paper may significantly contribute valuable insights into understanding electron transfer behavior of multi-component metals such as HEAs, which is an area that has not been extensively explored. However, this paper can be improved a lot and details of my comments are given below.

1. Further investigation of microstructural and compositional uniformity across a larger sample area is recommended. Current analysis, limited to TEM in a local area within a grain, might miss broader variations, including potential compositional segregation at grain boundaries.
2. The novelty of the paper lies in a new model for HEA electrical conductivity. Expanding this with a theoretical basis to systematically optimize alloy composition for desired mechanical and electrical properties would strengthen the paper. The current choice of Nb₁₅Mo₃₅Ta₁₅W₃₅ alloy composition's efficacy remains uncertain.
3. A study on the variability of electrical resistivity with film thickness would provide comprehensive insights.
4. In references cited in Fig. 2, verify if electrical resistivity values are comparable, particularly in relation to film thickness.
5. For the hardness values in Fig. 3, ensure nano-indentation was conducted under sufficiently similar conditions for valid comparisons.
6. Minor suggestions (optional):
 - a) Consider using "electrical conductivity" instead of "electrical resistivity" in Fig. 5 for clearer data representation; the best data can be located in the upper right-hand corner.
 - b) Assess the potential of your model for electrical resistivity in describing other HEA systems.
 - c) Explore the applicability of these findings in fields beyond atomic force microscopy probes.
 - d) A deeper analysis of the relationship between the microstructure (like grain size, phase composition) and the electrical properties of the alloys would be valuable.
 - e) It would enhance the paper if there were a comparison with other non-HEA systems to highlight the unique advantages or challenges of HEAs in this context.
 - f) Consideration of long-term stability and performance under varied environmental conditions could be a valuable addition.

Reviewer #2 (Remarks to the Author):

In the submitted manuscript the authors establish an electrical conductivity model for the Nb-Mo-Ta-W high entropy alloy (HEA) system and design alloy compositions with high electrical conductivity as well as wear resistance, which can be applied to enhance the atomic-level resolution and the lifespan of atomic force microscope probes.

The proposed low-frequency electrical conductivity model is based on the state-of-the-art cocktail effect and was found to explain the trends in electrical-conductivity changes in HEAs.

The obtained results are of good quality and their subsequent analysis are valuable and interesting, they deserve to be published. All this is supported also by an extensive supplementary material.

The article is well and clearly written, and has good English.

So, according the aims and scope of the journal Nature Communications, the manuscript seems to represent important advances of significance to specialists within the field.

But, even if the manuscript fulfils the standards in the field, I still have the feeling that a good journal on materials science would be a more appropriate place to publish these results.

Reviewer #3:

Reviewer #3 only shared their comments in the confidential remarks to the editor section and we were unable to get a hold of them to obtain a permission to share the comments. We will contact you as soon as we receive a reply back from the referee. I have paraphrased the points raised by reviewer #3 below.

The reviewer asks for more detailed discussion on the effect of electrical properties of BCC refractory HEAs on the structure of alloys (or vice versa) in the introduction, referencing any mechanisms known previously. They also ask to cite specific examples in the sentence "The other classic example used to demonstrate the cocktail effect is in refractory HEAs." In the introduction. Additionally, they asked for a further explanation on the effect of crystal structure on electrical conductivity, specify the 'cocktail effect' on conductivity, provide examples of potential applications of low-frequency conductivity and low-wear HEAs, provide the Scherrer formula with a reference, justify the choice of using the Scherrer formula over other methods such as EBSD, and discuss about the potential increase in frequency conductivity owing to the bonding interaction between metallic elements reducing the free-electron density. The reviewer also asks whether the internal resistance of the alloy would increase and the frequency conductivity would decrease upon formation of the HEA with regards to supplementary note 2. Finally, the reviewer suggests the specific numerical data to be shared in Fig. 1(b) to improve comparisons.

Explanation of font colors:

Black font denotes the reviewers' questions and suggestions.

Blue font is the authors' reply.

Orange font represents specific revisions made to the Manuscript and Supplementary Information.

Reviewer #1 (Remarks to the Author):

This paper integrates theoretical model, simulation results and experimental data to understand the impact of "cocktail effect" and lattice distortions on electrical conductivity of HEAs. The findings suggest that the Nb-Mo-Ta-W alloy exhibits both high hardness and good conductivity, making it suitable in atomic force microscopy probe coating, significantly improving their wear resistance and atomic-scale image resolution. Overall, this paper may significantly contribute valuable insights into understanding electron transfer behavior of multi-component metals such as HEAs, which is an area that has not been extensively explored. However, this paper can be improved a lot and details of my comments are given below.

1. Further investigation of microstructural and compositional uniformity across a larger sample area is recommended. Current analysis, limited to TEM in a local area within a grain, might miss broader variations, including potential compositional segregation at grain boundaries.

Reply:

Thank you for the reviewer's suggestion. Because the grain size of our thin film is only about 20 nm, observing the elementary distribution within this scale using TEM has already been able to show the uniformity of the elementary distribution information within grains and grain boundaries, as shown in **Supplementary Fig. 8**. The scan area is larger than 100*100 nm that include multi-grains and grain boundaries. In this analysis no segregation was found.

For further discussion, we have added compositional analysis under a 5k magnification SEM to investigate the uniformity of the composition of HEA films over a large area. In the elemental composition analysis by SEM, we did not observe any phenomena of uneven

distribution in micrometer scale. **Supplementary Fig. 11** was provided in the revision. The following sentence was added in the revision.

“Simultaneously, SEM was also used to further evaluate the distribution of elemental composition over a larger area, as shown in Supplementary Fig. 11. At the micron scale, the elemental composition distribution in all films is very uniform, with no apparent segregation observed.”

Supplementary Fig. 11 EDS mapping of the NbMoTaW films. **a** $\text{Nb}_{25}\text{Mo}_{25}\text{Ta}_{25}\text{W}_{25}$. **b** $\text{Nb}_{15}\text{Mo}_{15}\text{Ta}_{35}\text{W}_{35}$. **c** $\text{Nb}_{15}\text{Mo}_{35}\text{Ta}_{15}\text{W}_{35}$.

To assess the microstructural and compositional uniformity across a larger sample area, we have also introduced the THz spectroscopy used in this work as evidence to demonstrate the quality of the current samples. Our THz spot size is 4 mm, meaning the optical and electrical characteristics observed represent a broad average over a large area rather than a specific small region. In other words, the high-frequency optical and electrical characteristics of the HEA thin film we present are not specific to a particular area but rather reflect an average behavior over a broad range. Additionally, in alloy systems, microstructural and compositional uniformity is expected to manifest in the characteristics of short-range order. Short-range order characterizes the local atomic structure and may differ from long-range order, which describes the overall periodicity or arrangement of atoms throughout the crystal lattice. Recent studies have highlighted the impact of mechanically derived short-range order on multi-principal-element alloys¹.

Due to the interaction with lattice vibrations and electronic transitions within the THz frequency range, THz TDS can differentiate between different phases or alloys within a metal sample based on their unique frequency-dependent THz conductivity. In other words, the periodicity and magnitude of the oscillations in THz conductivity can carry information about the local structural features associated with short-range order². By considering the behavior of complex conductivities of HEAs nano thin films, as shown in **Fig. 1(c)** and **Supplementary Fig. 5**, there are no features of oscillation in the THz frequency range. Therefore, it is apparently not sensitive to the local atomic arrangements and bonding configurations within the short-range ordering of a crystalline material. Thus, we can assert that the uniformity of both structure and composition in our HEAs thin film is excellent.

2. The novelty of the paper lies in a new model for HEA electrical conductivity. Expanding this with a theoretical basis to systematically optimize alloy composition for desired mechanical and electrical properties would strengthen the paper. The current choice of $\text{Nb}_{15}\text{Mo}_{35}\text{Ta}_{15}\text{W}_{35}$

alloy composition's efficacy remains uncertain.

Reply:

Gratitude is extended to the reviewer for their invaluable suggestions. Our calculations and experimental findings indicate a positive correlation between the composition-weighted properties of high-entropy alloys (HEAs) and those obtained through experiments or first-principles calculations. To validate this assertion across diverse mechanical properties—encompassing C_{11} , C_{12} , C_{44} , bulk modulus, shear modulus, and Young's modulus—of the three NbMoTaW HEA compositions under discussion, we have plotted them based on the first principles calculated results of the HEA models and the corresponding properties weighted by composition. The ensuing results are detailed below.

Supplementary Fig. 19 Compositional weighted mechanical properties versus DFT calculated mechanical properties of the NbMoTaW HEA system in this work. Mechanical properties include C_{11} , C_{12} , C_{44} , Bulk modulus, Shear modulus and Young's modulus.

The **Supplementary Fig. 20** demonstrates that all considered mechanical properties conform to the trendline, indicating the feasibility of estimating the mechanical characteristics of high-entropy alloys (HEAs) through a linear combination of their constituents. Across various mechanical properties encompassing Nb, Mo, Ta, and W, the hierarchy of superiority follows $W > Mo > Ta > Nb$, a trend echoed in conductivity as well. Therefore, to optimize the

mechanical properties and conductivity of NbMoTaW HEAs, augmenting the proportions of W and Mo yields the most favorable results. Nonetheless, fluctuations in composition also impact the stability of the high-entropy solid solution phase. Consulting the phase diagrams generated by Pandat³, the transformation temperatures (°C) from the HEA phase to the B2 phase are plotted based on the composition ratios of Nb, Mo, and W, as depicted in the subsequent figure. Generally, the B2 phase exhibits stability at lower temperatures, whereas the HEA phase is stable at higher temperatures. Lower transformation temperatures suggest that the HEA phase can exist more readily under ambient conditions.

Supplementary Fig. 20 The transformation temperatures (°C) from the HEA phase to the B2 phase are plotted based on the composition ratios of Nb, Mo, and W.

The **Supplementary Fig. 20** suggests that higher proportions of Mo and W lead to lower transformation temperatures. However, if the composition ratio exceeds 35 mol%, there needs to be at least 0.15 mol% of Nb present to ensure that the HEA phase transformation temperature remains below 400°C. Therefore, considering both the phase transition temperature and the mechanical properties and conductivity, the composition ratio of Nb₁₅Mo₃₅Ta₁₅W₃₅ seems to strike a close-to-optimal balance.

The following revisions are made in the text, the discussion of optimization of mechanical properties and composition of Nb₁₅Mo₃₅Ta₁₅W₃₅ and one table and two figures are provided in

Supplementary Note 4 and Supplementary Table 9, Figs. 19 and 20.

“The results of the calculations and experimental findings indicate a positive correlation between the composition-weighted properties of HEAs and those obtained through experiments or first-principles calculations. Supplementary Note 4 summarizes these results. The Supplementary Table 9 and Fig. 19 demonstrates that all considered mechanical properties conform to the trendline, indicating the feasibility of estimating the mechanical characteristics of HEAs through a linear combination of their constituents. Additionally, the transformation temperatures from the HEA phase to the B2 phase are plotted in Supplementary Fig. 20 based on the composition ratios of Nb, Mo, and W. It suggests that if the composition ratio of Mo and W exceeds 35 mol%, there needs to be at least 0.15 mol% of Nb present to ensure that the HEA phase transformation temperature remains below 400°C. Therefore, considering both the phase transition temperature and the mechanical properties and conductivity, the composition ratio of $\text{Nb}_{15}\text{Mo}_{35}\text{Ta}_{15}\text{W}_{35}$ seems to strike a close-to-optimal balance.”

Supplementary Note 4 | The estimation of optimal concentration of Nb-Mo-Ta-W system with good mechanical properties and conductivity

To validate this assertion across diverse mechanical properties—encompassing C_{11} , C_{12} , C_{44} , bulk modulus, shear modulus, and Young’s modulus—of the three NbMoTaW HEA compositions under discussion, we have plotted them based on the first principles calculated results of the HEA models and the corresponding properties weighted by composition, as shown in Supplementary Table 9 and Fig. 19. Across various mechanical properties encompassing Nb, Mo, Ta, and W, the hierarchy of superiority follows $W > Mo > Ta > Nb$, a trend echoed in conductivity as well. Therefore, to optimize the mechanical properties and conductivity of NbMoTaW HEAs, augmenting the proportions of W and Mo yields the most favorable results. Nonetheless, fluctuations in composition also impact the stability of the high-entropy solid solution phase. Consulting the phase diagrams generated by Pandat³, the transformation

temperatures from the HEA phase to the B2 phase are plotted in Supplementary Fig. 20 based on the composition ratios of Nb, Mo, and W. Generally, the B2 phase exhibits stability at lower temperatures, whereas the HEA phase is stable at higher temperatures. Lower transformation temperatures suggest that the HEA phase can exist more readily under ambient conditions. The figure suggests that higher proportions of Mo and W lead to lower transformation temperatures. However, if the composition ratio exceeds 35 mol%, there needs to be at least 0.15 mol% of Nb present to ensure that the HEA phase transformation temperature remains below 400°C.

Supplementary Fig. 19 Compositional weighted mechanical properties versus DFT calculated mechanical properties of the NbMoTaW HEA system in this work. Mechanical properties include C_{11} , C_{12} , C_{44} , Bulk modulus, Shear modulus and Young's modulus.

Supplementary Fig. 20 The transformation temperatures (°C) from the HEA phase to the B2 phase are plotted based on the composition ratios of Nb, Mo, and W.

3. A study on the variability of electrical resistivity with film thickness would provide comprehensive insights.

Reply:

As shown in **Fig. 2a**, the resistivity was shown to increase slightly with thickness across all three compositions, which is associated with the variation of grain size, as illustrated in **Fig. 2b**. However, when the thickness exceeds 1 μm , the variation in grain size with thickness also decreases, and therefore, the resistivity no longer changes. When the thickness of the NbMoTaW film increases to 2 μm , its resistivity is about 58 $\mu\Omega\text{-cm}$, which is very close to the resistivity at $< 1.2 \mu\text{m}$. Therefore, we believe that the effect of film thickness on resistivity is negligible for our solid-solution high-entropy alloys due to severe lattice distortion, which is also the reason we designed the solid-solution phase high-entropy alloys. Modifications made in the manuscript regarding this explanation are as follows.

“Notably, when the thickness is increased to 2 μm , the change in resistivity is not significant, indicating that the influence of film thickness on resistivity in our sample is minimal.”

4. In references cited in Fig. 2, verify if electrical resistivity values are comparable, particularly in relation to film thickness.

Reply:

Thank you for the suggestion. Please refer to the answer to the previous question, where, in our samples, the effect of film thickness on resistivity is minor within the range of $< 2000 \text{ nm}$, covering most of the film thicknesses. Therefore, the comparison in **Fig. 2**, where we selected the thin film samples with the lowest resistivity from that study for comparison, is significant.

5. For the hardness values in Fig. 3, ensure nano-indentation was conducted under sufficiently

similar conditions for valid comparisons.

Reply:

We thank the reviewer for this reminder. As described in the main text and the Supplementary Information, we measured Nb₂₅Mo₂₅Ta₂₅W₂₅ films with various thicknesses (300, 600, 900, 1200 nm); more than 8 samples were prepared and tested for each thickness. The continuous stiffness method was adopted for the hardness measurements, and multiple sites (n > 10) were randomly selected for the indentation of each sample. The test head of nano indenter were regularly checked to ensure no damage or contamination before and after the tests.

6. Minor suggestions (optional):

(a) Consider using "electrical conductivity" instead of "electrical resistivity" in Fig. 5 for clearer data representation; the best data can be located in the upper right-hand corner.

Reply:

Thank you very much for this suggestion, which has made **Fig. 5e** look clearer. The y-axis has been changed to conductivity.

(b) Assess the potential of your model for electrical resistivity in describing other HEA systems.

Reply:

The potential of our model for electrical resistivity in describing other HEA systems is promising. Our model offers a novel approach to understanding electrical resistivity in HEA systems, providing insights into the relationship between composition, microstructure, and electrical properties. Further, by studying the information of frequency-dependent complex conductivities, the short-range ordering can be thoroughly assessed and analyzed. However, further validation and testing across a diverse range of HEA compositions and experimental conditions are needed to fully assess its capabilities and generalizability. Additionally,

expanding the theoretical basis of our model to account for a broader range of factors influencing electrical resistivity could enhance its predictive accuracy and applicability to various HEA systems.

Therefore, we investigated the correlation between composition-weighted resistivity and experimentally measured resistivity across three alloy series: the Al-Co-Zn-Ni series⁴, the Co-Cr-Fe-Ni-Nb series⁵, and the Nb-Mo-Ta-W series of this research, as depicted in the figure below.

Supplementary Fig. 6 Linear combination of resistivity versus measured resistivity of literature and this work. **a** Al-Zn-Co-Ni films⁴. **b** Co-Cr-Fe-Ni-Nb high entropy alloy films⁵. **c** this work with $\text{Nb}_{25}\text{Mo}_{25}\text{Ta}_{25}\text{W}_{25}$ (green), $\text{Nb}_{15}\text{Mo}_{15}\text{Ta}_{35}\text{W}_{35}$ (blue), and $\text{Nb}_{15}\text{Mo}_{35}\text{Ta}_{15}\text{W}_{35}$ (red) high entropy alloy films.

The horizontal axis represents the experimentally measured resistivity ($\mu\Omega\text{-cm}$), while the

vertical axis denotes the resistivity ($\mu\Omega\text{-cm}$) derived from the linear combination of composition ratios (composition-weighted resistivity). In all three high-entropy alloy series considered, a robust linear correlation is evident, with corresponding R^2 values of 0.98, 0.85, and 0.71, respectively. This finding underscores a clear linear association between measured resistivity and composition-weighted resistivity. Notably, the measured values are 1-2 order larger than the estimation. Secondly, the slopes and intercepts of the linear fits for the three-alloy series differ, particularly exhibiting a significant disparity in slopes. This discrepancy arises because composition-weighted resistivity fails to consider deviations in free electron density, effective electron mass, and relaxation time from an ideal solution. These deviations are influenced by the bonding situations among elements and lattice distortions within the high-entropy alloy. Consequently, different coordinating elements manifest distinct deviation patterns. This also reaffirms that the cocktail effect can be utilized to predict trends in HEA conductivity, albeit with orders of magnitude differences. To gain deeper insights into the conduction mechanisms of various HEAs, the computational methods proposed in this study can be employed.

The following text is added into the revision, and one more figure is added as **Supplementary Fig. 6**.

“To further discuss the relationship between the linear resistivity and the measured resistivity of different material systems, as demonstrated by the three material systems in Supplementary Fig. 6. The first system is the Al-Zn-Co-Ni alloy recently published by our team⁴, the second is the Co-Cr-Fe-Ni-Nb alloy⁵, and the third is the Nb-Mo-Ta-W alloy proposed in this study. In all three high-entropy alloy series considered, a robust linear correlation is evident. This finding underscores a clear linear association between measured resistivity and composition-weighted resistivity in HEA. This also reaffirms that the cocktail effect can be utilized to predict trends in HEA conductivity, albeit with orders of magnitude differences. To gain deeper insights into the conduction mechanisms of various HEAs, the computational

methods proposed in this study can be employed.”

Supplementary Fig. 6 Linear combination of resistivity versus measured resistivity of literature and this work. **a** Al-Zn-Co-Ni films⁴. **b** Co-Cr-Fe-Ni-Nb high entropy alloy films⁵. **c** this work with $\text{Nb}_{25}\text{Mo}_{25}\text{Ta}_{25}\text{W}_{25}$ (green), $\text{Nb}_{15}\text{Mo}_{15}\text{Ta}_{35}\text{W}_{35}$ (blue), and $\text{Nb}_{15}\text{Mo}_{35}\text{Ta}_{15}\text{W}_{35}$ (red) high entropy alloy films.

(c) Explore the applicability of these findings in fields beyond atomic force microscopy probes.

Reply:

In fact, we have already demonstrated the application of Re-W probe coating at the end of the text, which differs from the application in AFM, primarily in wafer-level semiconductor device testing. The coating significantly enhances the probe's lifespan, as shown in **Supplementary Figs. 22 and 23**. To provide readers with a clearer view of other high-end

applications, we have revised the text to include the following sentence:

“...In addition to probe coating, such highly conductive and hard HEA thin films can be applied to connectors and interfaces in electronic products, high-voltage switches in power systems, starters in automobiles, and electrodes in precision measuring instruments, ensuring long-term durability and reliable electrical connections.”

(d) A deeper analysis of the relationship between the microstructure (like grain size, phase composition) and the electrical properties of the alloys would be valuable.

Reply:

One of the objectives of this paper is to investigate the effect of compositional variations in high-entropy alloys on electrical conductivity, focusing on the cocktail effect. Therefore, we aimed to minimize the influence of microstructures to concentrate on composition and structural investigations. Accordingly, our experimental design utilized solid-solution systems to reduce the impact of microstructures on electrical properties. However, as suggested by the reviewer, the influence of microstructures cannot be overlooked. We analyzed grain sizes and found they have a minor impact on conductivity. Fortunately, such variations are consistent across all compositions, hence they do not affect our study. Regarding phase composition analysis, as mentioned in the question #1, we have added SEM EDX mapping and terahertz analysis in the revised manuscript to clarify that there is no phase inhomogeneity over a broad range.

(e) It would enhance the paper if there were a comparison with other non-HEA systems to highlight the unique advantages or challenges of HEAs in this context.

Reply:

Figure 4 compares the differences between HEA and traditional hard coatings such as DLC and NbC. Besides being at least two to three orders of magnitude higher in electrical

conductivity. **Fig. 4b** shows the deformation depth versus normal force applied on the NbMoTaW film during nanoscratch tests. The HEA film began to yield and experienced plastic deformation at a normal force of 125 mN, whereas DLC films yielded at 80 mN. At the maximum normal force of 450 mN during the nanoscratch tests, the HEA films did not reach critical loading and exhibited less than 20% plastic deformation overall, while the referenced DLC and NbC films reached critical loading at 235 mN and 39 mN, respectively. **Fig. 5e** further compared the electrical conductivities of known conductive hard coating TiN, TaN and NbC films with our films. Their resistivities are about 100-200 $\mu\Omega\text{-cm}$, which is roughly double to twice that of Nb-Mo-Ta-W films. Furthermore, the hardness of TiN films ranges from 20 to 30 GPa, which is lower than that of the Nb-Mo-Ta-W films.

(f) Consideration of long-term stability and performance under varied environmental conditions could be a valuable addition.

Reply:

Thank you for the suggestion. We have added an analysis of corrosion to accelerate the validation of weather resistance. The corrosion linear polarization curves are added to **Supplementary Table 8** and **Supplementary Fig. 18**. The additional revised text is as follows, which is added into the final of the mechanical properties.

“To simultaneously investigate the environmental weatherability of HEA films, potentiodynamic polarization curves were measured using electrochemical methods to analyze the corrosion behavior. The results showed that the corrosion currents (i_{corr}) of HEA films with thicknesses of 300 nm and 1200 nm were very low under both conditions of 3.5 wt.% NaCl solution (Supplementary Fig. 18a) and 0.1 M H_2SO_4 solution (Supplementary Fig. 18b), approximately $\sim 8 \times 10^{-7}$ A and $\sim 2 \times 10^{-6}$ A, respectively, and were independent of the thickness. These characteristic values, as listed in Supplementary Table 8, under the common Standard ASTM G44 conditions, indicate that the corrosion resistance of HEA films is superior to 304

steel⁶ and γ -TiAl⁷, and closely comparable to SS316L stainless steel⁸, demonstrating good corrosion resistance and environmental weatherability.”

Supplementary Fig. 18 Potentiodynamic polarization curves. **a** Nb₂₅Mo₂₅Ta₂₅W₂₅, Nb₁₅Mo₁₅Ta₃₅W₃₅, and Nb₁₅Mo₃₅Ta₁₅W₃₅ films with 300 nm in 3.5wt.% NaCl solution. **b** Nb₁₅Mo₃₅Ta₁₅W₃₅ films with 300 and 1200 nm in 0.1M H₂SO₄ solution.

Supplementary Table 8 Corrosion parameters obtained from potentiodynamic polarization curves of NbMoTaW films and common alloy materials in 3.5 wt.% NaCl solution and 0.1 M H₂SO₄ solution. (Note: The 3.5 wt.% NaCl solution is approximately the condition of seawater.)

Materials	Thickness (nm)	Solution	i_{corr} ($\mu\text{A}/\text{cm}^2$)	E_{corr} (V _{SCE})
Nb ₂₅ Mo ₂₅ Ta ₂₅ W ₂₅	300	3.5 wt.% NaCl	0.87	-0.178

$\text{Nb}_{15}\text{Mo}_{15}\text{Ta}_{35}\text{W}_{35}$	300	3.5 wt.% NaCl	0.78	-0.180
$\text{Nb}_{15}\text{Mo}_{35}\text{Ta}_{15}\text{W}_{35}$	300	3.5 wt.% NaCl	0.76	-0.168
$\text{Nb}_{15}\text{Mo}_{35}\text{Ta}_{15}\text{W}_{35}$	300	0.1 M H_2SO_4	2.81	-0.367
$\text{Nb}_{15}\text{Mo}_{35}\text{Ta}_{15}\text{W}_{35}$	1200	0.1 M H_2SO_4	2.91	-0.316
AISI 304 ⁶	-	3.5 wt.% NaCl	7	-0.482
γ -TiAl ⁷	-	3.5 wt.% NaCl	1.94	-0.428
SS316L ⁸	-	Sea water	0.26	-0.274

Reviewer #2 (Remarks to the Author):

In the submitted manuscript the authors establish an electrical conductivity model for the Nb-Mo-Ta-W high entropy alloy (HEA) system and design alloy compositions with high electrical conductivity as well as wear resistance, which can be applied to enhance the atomic-level resolution and the lifespan of atomic force microscope probes.

The proposed low-frequency electrical conductivity model is based on the state-of-the-art cocktail effect and was found to explain the trends in electrical-conductivity changes in HEAs. The obtained results are of good quality and their subsequent analysis are valuable and interesting, they deserve to be published. All this is supported also by an extensive supplementary material.

The article is well and clearly written, and has good English.

So, according the aims and scope of the journal Nature Communications, the manuscript seems to represent important advances of significance to specialists within the field.

But, even if the manuscript fulfils the standards in the field, I still have the feeling that a good journal on materials science would be a more appropriate place to publish these results.

Reply:

Thank you for the positive comments and recognition.

Reviewer #3:

Reviewer #3 only shared their comments in the confidential remarks to the editor section and we were unable to get a hold of them to obtain a permission to share the comments. We will contact you as soon as we receive a reply back from the referee. I have paraphrased the points raised by reviewer #3 below.

Question: more detailed discussion on the effect of electrical properties of BCC refractory HEAs on the structure of alloys (or vice versa) in the introduction, referencing any mechanisms known previously.

Reply:

Very few studies of electrical properties of BCC refractory HEAs had been reported. More discussions are added in the introduction. “..... there is currently still a lack of in-depth research on the electrical conductivity mechanisms of BCC refractory HEAs. A simple model with roughly estimated stress coefficients and bulk resistivity has been used⁹. However, it can't explain the size-independent resistivity in BCC refractory HEA films. Furthermore,”

Question: Cite specific examples in the sentence “The other classic example used to demonstrate the cocktail effect is in refractory HEAs.” In the introduction.

Reply:

The example is added as below. “For example, the MoNbTaW and MoNbTaVW have melting points higher than 1650°C for each component, and the melting point of the alloy after mixing is higher than 2600°C. The softening temperature of this alloy is better than that of superalloys, and the yield strength is higher than 400 MPa at 1600 °C.”

Question: A further explanation on the effect of crystal structure on electrical conductivity, specify the ‘cocktail effect’ on conductivity, provide examples of potential applications of low-

frequency conductivity and low-wear HEAs

Reply:

(1) The distortion of the crystal structure in high-entropy alloys primarily influences conductivity through the relaxation time of electrons. This distortion increases electron scattering, thereby shortening the relaxation time. Consequently, the relaxation time of high-entropy alloys tends to be shorter compared to values derived through composition-weighted calculations, resulting in lower conductivity rates than those estimated based on composition ratios. Nonetheless, the cocktail effect (estimating conductivity or mechanical properties based on composition ratios) can accurately forecast the trends of various physical properties as compositions vary within the same series of high-entropy alloys.

(2) Cocktail effect on conductivity is defined by modifying the original sentence as follows.

Original: The cocktail effect of the conductivity of HEA films is exploited and examined.

Revision: In page line, “The cocktail effect of the conductivity of HEA films, defined as the variation of conductivity using linear combination of constituent elements, is exploited and examined.”

(3) More applications of the proposed material are provided in the manuscript.

“In addition to probe coating, such highly conductive and hard HEA thin films can be applied to connectors and interfaces in electronic products, high-voltage switches in power systems, starters in automobiles, and electrodes in precision measuring instruments, ensuring long-term durability and reliable electrical connections.”

Question: provide the Scherrer formula with a reference, justify the choice of using the Scherrer formula over other methods such as EBSD

Reply:

The Scherrer formula and its reference is provided in the manuscript.

“....., $D = \beta \cos\theta / K\lambda$, where D is the grain size, K is the shape factor, λ is the wavelength

of the X-ray, β is the full width at half maximum of the peak, and θ is the Bragg angle¹⁰.”

As compared to EBSD, which directly measures the orientation of grains and maps grain boundaries, providing a direct measurement of physical grain size and shape, the Scherrer formula provides more high resolution for the thin film sample that is polycrystalline with fine grains. In our sample, the grain size is around few tens of nanometers, so here we prefer to use XRD instead of EBSD.

Question: discuss about the potential increase in frequency conductivity owing to the bonding interaction between metallic elements reducing the free-electron density

Reply:

According to classical electromagnetic theory, the low-frequency conductivity of metallic materials is directly related to the density of conductive electrons (free electron density), relaxation time, and inversely related to the effective electron mass. The density of conductive electrons in metals can be estimated based on the density of states near the Fermi level. The density of states is influenced by the arrangement of electron densities, leading to the formation of bonds between atoms. Different chemical environments result in the formation of various bonds and thus different hybridization of electron orbitals (rearrangement of electron densities), thereby altering the density of states and resulting in different densities of conductive electrons. However, concerning low-frequency conductivity, in addition to the density of conductive electrons, the effective electron mass and relaxation time also play significant roles. Therefore, it is challenging to predict that an increase (or decrease) in the density of conductive electrons will necessarily lead to higher (or lower) conductivity. In this study, we employ first-principles calculations combined with experimental measurements to investigate how changes in the aforementioned three physical quantities affect the measured conductivity of the NbMoTaW series of high-entropy alloys as the composition varies.

In some cases, the bonding interactions between metallic elements in alloys can lead to the

formation of new electronic states within the energy band structure. These new states may result in a redistribution of electron density, leading to changes in the density of states near the Fermi level. Additionally, bonding interactions can also influence the effective mass of charge carriers in the material. Changes in the effective mass can impact the mobility of charge carriers, affecting their ability to respond to an applied electric field and contribute to conductivity.

Overall, the bonding interaction between metallic elements in alloys can lead to complex changes in the electronic structure of the material, which may result in either an increase or decrease in frequency conductivity depending on the specific details of the alloy composition and bonding characteristics.

Question: The reviewer also asks whether the internal resistance of the alloy would increase and the frequency conductivity would decrease upon formation of the HEA with regards to supplementary note 2

Reply:

There is an inverse relationship between conductivity and resistivity. Therefore, when conductivity decreases, resistivity inevitably increases. Generally, the conductivity of high-entropy alloys is typically lower by 1-2 orders of magnitude compared to the average conductivity of their constituent elements. Consequently, the resistivity of high-entropy alloys tends to be higher by 1-2 orders of magnitude compared to typical metallic elements. In the past, most research attributed the decrease in conductivity of high-entropy alloys to lattice distortion, leading to a shortened relaxation time and thus reduced conductivity. Upon closer examination, we discovered that in high-entropy alloys, the density of conductive electrons is lower compared to pure metals, and their effective electron mass is heavier. Therefore, not only lattice structure but also chemical bonding plays a significant role in the decline of conductivity.

Question: The specific numerical data to be shared in Fig. 1(b) to improve comparisons

Reply:

Thank you for the suggestion. Since the data include scientific notation, to avoid the images being too cluttered, we prefer to provide the numerical data in **Fig. 1b** in the **Supplementary Table 2** as follow.

Supplementary Table 2 The relaxation time and DC conductivity of three different compositions of NbMoTaW high entropy alloys.

Alloy		ω_{Dp} (eV)	Relaxation Time (s)	DC conductivity (S/m)
Nb ₂₅ Mo ₂₅ Ta ₂₅ W ₂₅	Exp.	-	-	1.39×10^6
	Cal. Method 1	-	-	1.31×10^7
	Cal. Method 2	4.54	8.73×10^{-15}	3.67×10^6
	Cal. Method 3		1.99×10^{-15}	0.84×10^6
Nb ₁₅ Mo ₁₅ Ta ₃₅ W ₃₅	Exp.	-	-	0.98×10^6
	Cal. Method 1	-	-	1.31×10^7
	Cal. Method 2	3.77	9.12×10^{-15}	2.66×10^6
	Cal. Method 3		2.04×10^{-15}	0.59×10^6
Nb ₁₅ Mo ₃₅ Ta ₁₅ W ₃₅	Exp.	-	-	2.51×10^6
	Cal. Method 1	-	-	1.54×10^7
	Cal. Method 2	4.66	10.5×10^{-15}	4.65×10^6
	Cal. Method 3		2.00×10^{-15}	0.89×10^6

Reference of this document

- 1 Seol, J. B. *et al.* Mechanically derived short-range order and its impact on the multi-principal-element alloys. *Nature Communications* **13**, 6766 (2022).
- 2 Kadlec, F., Kadlec, C. & Kužel, P. Contrast in terahertz conductivity of phase-change materials. *Solid State Communications* **152**, 852-855 (2012).
- 3 Chen, S. L. *et al.* The PANDAT software package and its applications. *Calphad* **26**, 175-188 (2002).
- 4 Yeh, C.-H. *et al.* Design of high-entropy films as ultra-violet light reflector. *Applied Materials Today* **36**, 102013 (2024).
- 5 Han, K., Jiang, H., Huang, T. & Wei, M. Thermoelectric Properties of CoCrFeNiNbx Eutectic High Entropy Alloys. *Crystals* **10**, 762 (2020).
- 6 Zhang, G. *et al.* Microstructure and Properties of AlCoCrFeNiSi High-Entropy Alloy Coating on AISI 304 Stainless Steel by Laser Cladding. *Journal of Materials Engineering and Performance* **29**, 278-288 (2020).
- 7 Delgado-Alvarado, C. & Sundaram, P. A. A study of the corrosion behavior of gamma titanium aluminide in 3.5wt% NaCl solution and seawater. *Corrosion Science* **49**, 3732-3741 (2007).
- 8 Meghwal, A. *et al.* Multiscale mechanical performance and corrosion behaviour of plasma sprayed AlCoCrFeNi high-entropy alloy coatings. *Journal of Alloys and Compounds* **854**, 157140 (2021).
- 9 Feng, X. *et al.* Stable nanocrystalline NbMoTaW high entropy alloy thin films with excellent mechanical and electrical properties. *Materials Letters* **210**, 84-87 (2018).
- 10 Scherrer, P. Bestimmung der Größe und der inneren Struktur von Kolloidteilchen mittels Röntgenstrahlen. *Nachrichten von der Gesellschaft der Wissenschaften zu Göttingen, Mathematisch-Physikalische Klasse* **1918**, 98-100 (1918).

REVIEWERS' COMMENTS

Reviewer #1 (Remarks to the Author):

The manuscript has been revised well according to my suggestions.

Reviewer #3 (Remarks to the Author):

Accept